# Mechanistic?

**Naomi Saphra**[*]
The Kempner Institute at Harvard University
nsaphra@fas.harvard.edu

**Sarah Wiegreffe**[*]
Ai2 & University of Washington
wiegreffesarah@gmail.com

## Abstract

The rise of the term *mechanistic interpretability* has accompanied increasing interest in understanding neural models—particularly language models. However, this jargon has also led to a fair amount of confusion. So, what does it mean to be *mechanistic*? We describe four uses of the term in interpretability research. The most narrow technical definition requires a claim of causality, while a broader technical definition allows for any exploration of a model's internals. However, the term also has a narrow cultural definition describing a cultural movement. To understand this semantic drift, we present a history of the NLP interpretability community and the formation of the separate, parallel *mechanistic* interpretability community. Finally, we discuss the broad cultural definition—encompassing the entire field of interpretability—and why the traditional NLP interpretability community has come to embrace it. We argue that the polysemy of *mechanistic* is the product of a critical divide within the interpretability community.

## 1 Introduction

The field of mechanistic interpretability is growing dramatically, constantly motivating new workshops, forums, and guides. And yet, many are unsure what the term *mechanistic interpretability* entails. Researchers, whether experienced or new to the field, often ask what makes some interpretability research "mechanistic" (Andreas, 2024; Beniach, 2024; Hanna, 2024; Belinkov et al., 2023). Because both fields study language models (LMs), the distinction between traditional NLP interpretability (NLPI) and mechanistic interpretability (MI) is unclear. In fact, when work is labelled as *mechanistic* interpretability research, the label may refer to:

1. **Narrow technical definition:** A technical approach to understanding neural networks through their causal mechanisms.

2. **Broad technical definition:** Any research that describes the internals of a model, including its activations or weights.

3. **Narrow cultural definition:** Any research originating from the MI community.

4. **Broad cultural definition:** Any research in the field of AI—especially LM—interpretability.

Exacerbating this confusion, *mechanistic interpretability* in the narrow cultural definition describes the authors of a paper, rather than their methods or objectives. We must therefore discuss the landscape of the interpretability community itself in order to clarify the usage of *mechanistic interpretability*.

To that end, we will begin by characterizing the *narrow technical definition* (§2.1) and subsequently explain how the coinage of the term *mechanistic interpretability* led inevitably to its *broad technical definition* (§2.2). Both technical definitions characterize subsets of research from the NLPI community, but their work is not always classified as MI, illustrating the term's contextual application.

In order to understand how semantic drift eventually gave rise to the cultural definitions, we overview the history of the two distinct communities (NLPI and MI) (§3). We describe how a new movement of AI safety researchers, motivated by philosophical arguments for the importance of interpretability, differentiated themselves as the MI community in its *narrow cultural definition* (§3.2). The resulting financial and social context of the field now incentivizes NLPI researchers to bridge this gap by embracing the label in its *broad cultural definition* (§3.3).

*Mechanistic* is just one example of the imprecise and ambiguous language used in interpretability research. Although clarity is key for distilling and communicating insights about neural networks, we

---

[*] Equal contribution. Order chosen for aesthetics.

compare it to a number of similarly vague terms in the history of NLPI (Appendix A). However, in contrast with other cases of lexical ambiguity in the area, we argue *mechanistic* is notable because it exposes a cultural divide—one which is worth bridging for the sake of scientific progress.

## 2 So what is *mechanistic*?

Before the term *mechanistic* described a cultural movement, NLPI researchers occasionally used the term *mechanisms* to refer to internal algorithmic implementation (Belinkov, 2018), as suggested by Marr's levels of analysis (Marr and Poggio, 1976). The earliest uses of *mechanistic interpretability* also draw on this technical meaning, as do most current explicit definitions of the term. What, then, is this precise technical meaning?

### 2.1 From causality and psychology to NLP

Mechanistic interpretability derives its name from *causal mechanisms*. In a causal model, a causal mechanism is a function—governed by "lawlike regularities" (Little, 2004)—that transforms some subset of model variables (causes) into another subset (outcomes or effects). Causal mechanisms are a necessary component of any causal model explaining an outcome (Halpern and Pearl, 2005a,b).

The *narrow technical definition* of MI thus describes research that discovers causal mechanisms explaining all or some part of the change from neural network input to output at the level of intermediate model representations. For example, one mechanistic interpretation explains how an LM can predict "B" from the input sequence "ABABA" using *induction heads* (Olsson et al., 2022): attention heads that search for a previous occurrence of "A" in combination with other heads that attend to the token that follows it. This narrow definition of MI requires causal methods of understanding, but excludes those that do not investigate intermediate neural representations, such as behavioral testing with input-output pairs (e.g., Ribeiro et al., 2020; Xie et al., 2022). It also excludes non-causal methods, such as describing representational structure or correlating activation features with particular inputs and outputs.

Psychology and philosophy have long stressed the importance of causal mechanisms in explanations. Psychology studies show (Legare and Lombrozo, 2014; Vasilyeva and Lombrozo, 2015) that humans prefer explanations containing causal

mechanisms underlying an event over Aristotle's other modes (Lombrozo, 2016) of explanation. Tan (2022) argues that likewise, explanations in machine learning should focus on causal mechanisms linking input and output. Real-world causal models are complex and have many possible pathways to outcomes (Hesslow, 1988); complete explanations of such models would be burdensome and counterproductive. Therefore, explanation requires distillation (Jacovi and Goldberg, 2021). Human explanations distill by capturing only *proximal* mechanisms (Lombrozo, 2006)—those which are closest to, or immediately responsible for, the outcome.

Unlike the human brain, neural networks can be rigorously studied due to our ability to perform causal interventions on them. Because we are not limited to proximal mechanisms in our efforts to discover causal mechanisms in neural networks, we instead rely on *causal abstraction* (Beckers and Halpern, 2019; Beckers et al., 2020) for distillation: the theory that causal models at higher levels of granularity can be faithful simplifications of the true causal model, and thus serve as mechanistic interpretations of the network (Geiger et al., 2024a).

In an attempt to bring definitional rigor to MI, recent work in causal interpretability of neural networks has advocated for an even narrower technical definition of MI: explanation through a *complete end-to-end causal pathway from model inputs to outputs* via intermediate neural representations (Geiger et al., 2021, 2024b; Mueller et al., 2024). This definition excludes most early work in MI (§2.2), and has not yet been widely adopted. Induction heads, for example, only describe one component in the causal pathway—under the end-to-end definition, one would also need to explain how the model identifies the input "ABABA" as a 2-token repeating pattern, and then how the model predicts "B" after attending to earlier occurrences of it.

### 2.2 The coinage of *mechanistic interpretability*

The term *mechanistic interpretability* was coined by Chris Olah and first publicly used in the Distill.pub **Circuits thread**, a series of blogposts by OpenAI researchers between March 2020–April 2021. The first post (Olah et al., 2020) set out to "understand the *mechanistic* implementations of neurons in terms of their weights." After researchers involved in the Circuits thread moved to Anthropic, their subsequent reports (the **Transformer Circuits thread;** Elhage et al., 2021; Olsson et al., 2022; Hernandez et al., 2022; Elhage

et al., 2022a) leaned into the terminology heavily and eventually it became mainstream.

Elhage et al. (2021) provided the first explicit definition of MI: "attempting to *reverse engineer the detailed computations* performed by Transformers, similar to how a programmer might try to reverse engineer complicated binaries into human-readable source code." The analogy to reverse engineering, building on Olah (2015)'s earlier comparisons to code via functional programming, has had staying power. Recent definitions, such as that of the ICML 2024 MI workshop (Barez et al., 2024), use similar wording:

> "...reverse engineering the algorithms implemented by neural networks into human-understandable mechanisms, often by examining the weights and activations of neural networks to identify circuits ...that implement particular behaviors."

While this definition still implicitly focuses on causal mechanisms (the key technical distinction one can draw between MI and some other subtypes of interpretability), current MI research rarely makes reference to causality. Reflecting both the emphasis above on examining weights and activations and the definition's lack of specificity about acceptable methods, many recent works have adopted a *broad technical definition* of MI to mean *any* inspection of model internals.[1] This semantic drift may have been inevitable—how could we reverse engineer a network without first inspecting its internal components? However, the further generalization of the term to label the output of a community, rather than its characteristic approach, was perhaps less inevitable.

## 3 How did we get here? A history of two LM interpretability communities

Our current terminological confusion results from a historical[2] accident: MI started as a movement with distinct technical objectives in computer vision, but ultimately moved into NLP without engaging the existing community which was already pursuing similar objectives.[3]

### 3.1 The nascent field of NLP Interpretability

NLP researchers published focused analyses of linguistic structure in neural models as early as 2016, primarily studying recurrent architectures like LSTMs (Ettinger et al., 2016; Linzen et al., 2016; Li et al., 2016; Hupkes et al., 2017; Ding et al., 2017). The growth of the field, however, also coincided with the adoption of Transformers, which were initially developed for machine translation and constituent parsing (Vaswani et al., 2017) but rapidly dominated rankings across standard NLP tasks (Radford et al., 2018, 2019; Devlin et al., 2019), drawing wide interest in understanding how they worked.

To serve the expanding NLPI community, the first BlackBoxNLP workshop (Alishahi et al., 2019) was held in 2018 and immediately became one of the most popular workshops at any ACL conference. Whereas many NLPI researchers had previously struggled to find an audience, ACL implemented an "Interpretability and Analysis" main conference track in 2020 (Lawrence, 2020), reflecting the mainstream success of the field.

In many ways, the early NLPI field—which related model behavior to particular components, layers, and geometric properties—would be familiar to anyone in the current MI community. Not only is current research often reinventing their methods and rediscovering their findings (§3.2.2), it is also repeating the same epistemological debates. These debates pit correlation against causation, simple features against complex subnetworks, and expressive mappings against constrained interpretations.

### 3.1.1 Distributional semantics and representational similarity

Interest in vector semantics exploded in the NLP community after word2vec (Mikolov et al., 2013a) popularized many approaches to interpreting word embeddings (Mikolov et al., 2013b,c).[4] The en-

---

[1]The minimal overlap between causality and MI has been previously noted (Mueller, 2024; Mueller et al., 2024).

[2]Note that our "history" turns on events barely two years before the time of writing. We are not overreaching, however, by assuming that many new researchers in this rapidly growing field are unfamiliar with its history. Popular MI tutorials and guides often begin their LM literature review in 2021-2022 (e.g., Docker and Nanda, 2023; Li, 2024; Nanda, 2024b), providing a limited window for many new entrants to the field.

[3]Here, we discuss MI and NLPI work under the *narrow cultural definition*. Although some of these MI papers fall outside of the *technical definitions*, most either self-label as MI or appear in MI venues. Regardless, not all of it is referred to as MI by the authors themselves, who may be more prescriptive in their own definitions. Our categorization of culture is based on the authors' networks and background: A paper's lead authors are MI if they entered the field through the MI or associated alignment community and NLPI if they are closely tied to the ACL interpretability community.

[4]These methods, first introduced in distributional semantics (Louwerse and Zwaan, 2009; Jurgens et al., 2012; Turney,

thusiasm for unigram embedding analysis proved transient, but still influences neural interpretability methods (Ethayarajh, 2019; Reif et al., 2019; Hernandez and Andreas, 2021). Distributional semantics has generalized to representational similarity methods (Saphra and Lopez, 2019b; Raghu et al., 2017; Wu et al., 2020) and vector space analogical reasoning has left clear marks on methods like task vectors (Ilharco et al., 2023) and steering vectors (Subramani et al., 2022; Turner et al., 2023). Many works in MI similarly leverage additive properties in representations (Marks and Tegmark, 2024; Tigges et al., 2023; Arditi et al., 2024).

Despite the brief excitement around distributional semantics, critics quickly noted that not all interesting properties of word embeddings correlated with downstream model behavior (RepEval, 2016). Furthermore, geometric analysis revealed these representations to be anisotropic and heavily influenced by word frequency rather than meaning (Mimno and Thompson, 2017). These critiques remain salient to modern correlational interpretability methods, including similarity-based metrics (Davari et al., 2023).

### 3.1.2 Attention maps

Attention, originally developed for recurrent machine translation models (Bahdanau et al., 2015), was rapidly adopted across language tasks. Even before the switch to fully attentional Transformers, attention modules offered new avenues of explanation (Bahdanau et al., 2015; Wang et al., 2016). In BERT models, the concurrent discovery of both a correlational (Clark et al., 2019; Htut et al., 2019) and causal (Voita et al., 2019) relationship between syntax and attention demonstrated the case for attention maps as a window into how Transformer LMs handled complex linguistic structure. However, NLPI researchers also identified some limitations of attention for interpretability (Serrano and Smith, 2019; Jain and Wallace, 2019; Wiegreffe and Pinter, 2019; Bibal et al., 2022). Some issues have longstanding solutions, such as incorporating the context of the model when computing attention metrics (Brunner et al., 2020; Kobayashi et al., 2020; Abnar and Zuidema, 2020).

MI work has continued to attribute specific stereotyped behavior to attention heads (Olsson

et al., 2022) and to present attention patterns as input saliency maps (Wang et al., 2023; Lieberum et al., 2023; Hanna et al., 2023), though more frequently with results that are causally validated.

### 3.1.3 Neuron analysis and localization

Early works on localizing concepts in NLP often associated individual neurons with sentiment, syntax, bias, or specific token sequences (Radford et al., 2017; Na et al., 2019; Bau et al., 2019; Lakretz et al., 2019; Dalvi et al., 2019; Durrani et al., 2020). Many such studies validated their findings by using causal interventions, though few proposals were causal by design (Sajjad et al., 2022). MI research has largely pursued similar goals of localizing model behaviors to fine-grained model components, including neurons, through its focus on finding "circuits": groups of components forming a sub-network that closely (or faithfully) replicate the full model's performance on a fine-grained task (Olah et al., 2020; Wang et al., 2023).

Single neuron analysis has been subject to criticism arguing that it is infeasible to reduce large, complex models to the sum of their parts (Antverg and Belinkov, 2022; Sajjad et al., 2022). One core problem is polysemanticity: the observation that a single neuron can activate for multiple disparate classes or concepts (Olah et al., 2020; Mu and Andreas, 2020; Bolukbasi et al., 2021). Not only are these concepts ambiguous, but they can combine nonlinearly according to a sequence's underlying latent structure (Saphra and Lopez, 2020; Csordás et al., 2024), making them difficult to disentangle and isolate. MI struggles with many of the same neuron analysis concerns as earlier work, but has taken a particular interest in resolving polysemanticity (Elhage et al., 2022b; Gao et al., 2024). One popular method for this purpose, the sparse autoencoder (SAE) (Bricken et al., 2023; Cunningham et al., 2024), still relies on assumptions of linearity (Park et al., 2024; Millidge, 2023) and naturally emerging feature sparsity (Saphra and Lopez, 2019a; Puccetti et al., 2022; Elhage et al., 2023). Like earlier neuron analysis methods, it also requires expensive causal validation (Mueller et al., 2024).

### 3.1.4 Component analysis and probing

Probes were first applied in NLPI to extract linguistic information from the hidden states of neural models (Ettinger et al., 2016; Kádár et al., 2017; Shi et al., 2016; Adi et al., 2017; Hupkes et al.,

---

2012), had previously relied on Latent Semantic Allocation (Turney, 2005) or other word representations derived from matrix factorization—a class that also, implicitly, includes word2vec itself (Levy and Goldberg, 2014).

2017; Belinkov et al., 2017a,b; Giulianelli et al., 2018). Many early papers observed that lower layers encode local features, echoing findings in computer vision (Yosinski et al., 2014) and reflecting the classical NLP pipeline (Tenney et al., 2019).

The probing literature quickly came under scrutiny (Belinkov, 2022) for its lack of proper baselines (Hewitt and Liang, 2019) or informative structural constraints (Saphra and Lopez, 2019b). Without proper experiment design, many probing methods appeared to extract more information from random embeddings than from trained representations (Zhang and Bowman, 2018; Wieting and Kiela, 2019). To manage these issues, newer probes incorporated information complexity (Pimentel et al., 2020; Voita and Titov, 2020) or used simple geometric mappings (Hewitt and Manning, 2019; White et al., 2021). Some designs reflected the model's own processing (Pimentel et al., 2022), as now exemplified by methods like the logit lens (nostalgebraist, 2020; Geva et al., 2022; Chuang et al., 2024) used widely in MI research. However, the logit lens—like other probing methods before it (Belinkov, 2022)— has been criticized for providing a largely incomplete causal explanation (Nanda, 2024b; Zhu et al., 2024; Wiegreffe et al., 2024).

Recent MI work has focused on projecting LM hidden states to interpretable subspaces using linear probes. These probes may be supervised (Li et al., 2023; Marks and Tegmark, 2024) or unsupervised, using an SAE. These methods inherit many critiques from the classic probing literature, including a lack of causal grounding. Recent proposals have therefore argued for validation by causal interventions for SAEs (Mueller et al., 2024), echoing previous efforts to validate probed structures (Giulianelli et al., 2018; Elazar et al., 2021).

## 3.2 The beginnings of mechanistic interpretability

As NLPI researchers continued investigating language model features and weights, their community and scientific understanding grew rapidly. However, they could not have predicted how the field would grow and change with the infusion of MI researchers into the area. To fully understand the lexical landscape of the NLPI field, we must consider how *mechanistic* historically came to denote a cultural split from the previous NLPI community in the term's *narrow cultural definition*.

### 3.2.1 The historical context of *mechanistic*
Though it may be surprising in the modern era of LLM hype, not long ago "machine learning" referred primarily to computer vision research. When Saphra (2021) analyzed the proceedings of ICML 2020, they found that over three times as many papers referenced CVPR as any *ACL conference, demonstrating that the language modality was relegated to an application while computer vision results were seen as core machine learning.

The presumed unmarked nature of image classification research shaped the landscape of interpretability research as well: In computer vision work at the time, the dominant interpretability method was gradient-based saliency, which highlighted the importance of specific pixels in an input image (Simonyan et al., 2014; Bach et al., 2015; Springenberg et al., 2015; Zintgraf et al., 2017; Ribeiro et al., 2016; Shrikumar et al., 2017). Meanwhile, NLP researchers (and other ML researchers experimenting on text) occasionally borrowed saliency methods from computer vision (Karpathy et al., 2016; Li et al., 2016; Arras et al., 2016; Lei et al., 2016; Alvarez-Melis and Jaakkola, 2017), but primarily sought to understand models through representational geometry, attention maps, probing, and causal or correlational neuron analysis—all methods employed by the MI community today.

When Chris Olah first described "mechanistic interpretability" in 2020, then, this was the cultural landscape of the ML field: Machine learning mostly meant image classification and interpretability mostly meant feature saliency. Olah has confirmed on multiple occasions (Olah, 2024a,b) that he coined the term to differentiate circuit analysis from saliency methods, which were subject to increasing skepticism at the time (Kindermans et al., 2016; Adebayo et al., 2018; Kindermans et al., 2019; Ghorbani et al., 2019; Heo et al., 2019; Slack et al., 2020; Zhang et al., 2020). The MI paradigm was crucial and novel within computer vision—but the community around it didn't stay in computer vision.

### 3.2.2 Two LM interpretability communities
As excitement grew around new breakthroughs in NLP and dialogue systems, particularly with the rise of powerful Transformer language models such as GPT-3+ (Brown et al., 2020), many researchers migrated domains. The Circuits thread itself changed focus from vision to language in

2021 (Elhage et al., 2021), with the subsequent discovery of induction heads (Olsson et al., 2022) moving beyond existing efforts to characterize individual predictable attention heads (Kovaleva et al., 2019) to instead interpret the interaction between pairs of such heads. These new discoveries excited the NLPI community, but—unlike in computer vision—MI's goals and methods represented a direct continuation of the existing field.

Instead of a difference in methodology, the MI community brought a distinct *culture* to LM analysis. They came from outside of NLP or even from outside of ML entirely, often drawn by arguments that LMs posed an existential risk which could be tempered by deeper understanding.[5] Until mid-2023, most MI research was shared on blogs or forums such as LessWrong and The AI Alignment Forum; on Discord [6] and Slack [7]; or at invite-only workshops (MIT, 2023). Findings were rarely published on arXiv or in academic venues— and some members of the alignment community even advocated against publication, arguing that it would advance dangerous AI capabilities (Hobbhahn and Chan, 2023), though others advocated for more engagement with academics (@typedfemale, 2023). While MI researchers may have taken NLPI researchers' absence in online forums as a sign that they were uninterested in MI, many NLPI researchers were unaware of the MI community growing outside traditional research and publication venues.

As the MI community expanded and largely switched focus to language models, technical distinctions became less important than these cultural differences. In his popular guide to the field, Nanda (2022, ref. "The Broader Interpretability Field") avoided a strict technical definition of mechanistic interpretability, instead stating it "*feels* distinct," differentiated by its "culture," "research taste," and epistemics. Attempts to differentiate mechanistic from non-mechanistic interpretability quickly became untenable, leading to incongruent ontologies. For example, Nanda (2022) categorized *activation patching*—which Nanda attributed to the ROME paper (Meng et al., 2022)[8]—as MI but ROME

itself—which uses activation patching to perform model editing—as non-MI. The modern MI community has even abandoned the early definitional goal of distinguishing MI from saliency—gradient-based feature attribution has re-emerged as another tool in the MI toolbox (Nanda, 2023a; Kramár et al., 2024).

To whatever degree *mechanistic* originally reflected a formal notion of causal mechanisms (§2.2), few researchers retain such a strict definition today. Instead, the formation of a separate, parallel language model interpretability community has led the term to its *narrow cultural definition*.

### 3.2.3 The clash of communities

The MI community eventually began publishing in academic conferences (Nanda, 2023b; Nanda et al., 2023; Wang et al., 2023). However, new engagement with academia only served to highlight bifurcated norms in the field. Researchers in the NLPI community expressed frustration on social media with the MI community's unfamiliarity with LM interpretability work prior to Anthropic's 2021 Circuits thread. Belinkov (2023a) argued that one paper "fail[ed] to engage with a large body of work on these topics from the past ~5 years," including direct precedents and improved baselines. Saxon (2023) alluded to a "contingent of people studying LLMs [who] don't meaningfully engage with *ACL literature." Others publicly stated that specific work from the MI community was "not new" (Artzi, 2023) or "published in the past" (Ravfogel, 2023). Posts often highlighted a tendency to "reinvent" (Andreas, 2023) or "rediscover" (Davidson, 2024) existing tools.

And yet, despite these tensions, the energy and resources of the growing MI community could not be denied. Many NLPI researchers subsequently began to use the term *mechanistic interpretability* to signal their engagement with the MI conversation (Nanda, 2024a).

### 3.3 We are all mechanistic now

Who wouldn't want to work on mechanistic interpretability? Students need advisors.[9] Funders

---

[5]Since 2021, the ML Alignment & Theory Scholars program (MATS), supported by the Berkeley Existential Research Initiative, has become a key point of entry for new researchers entering the interpretability field from outside of machine learning.
[6]e.g., https://mechinterp.com/reading-group
[7]e.g., https://opensourcemechanistic.slack.com
[8]Although the technique was first applied to neural net-

works by Vig et al. (2020) and Geiger et al. (2020).
[9]Prof. Sasha Rush of Cornell Tech noted, "pre-PhD researchers...[are] most excited about...'mechanistic interpretability'" (Rush, 2024).

need grant recipients.[10] There is free pizza.[11] Is it any surprise that the traditional NLPI community increasingly embraces the term? Because the MI community accounts for much of the current growth in interpretability research (Räuker et al., 2023), the term has ceased to distinguish two separate communities and has grown into its *broad cultural definition*, encompassing the work of all interpretability researchers.

Simply embracing the term, however, has not fully unified these communities. Although MI forum posts are often methodologically similar to papers at ACL, some differences persist.[12] The traditional NLPI community tends, for example, to be interested in using linguistics (Sarti et al., 2024; Mohebbi et al., 2023; Katinskaia and Yangarber, 2024) and automata theory (Weiss et al., 2018, 2021; Merrill et al., 2022, 2020, 2024) as analytic tools. These topics are niche—but growing—in MI.

The MI community has its own characteristic interests, such as training dynamics (Olsson et al., 2022; Liu et al., 2023; Nanda et al., 2023; Zhong et al., 2023)—though the NLPI community has also studied this topic (Chiang et al., 2020; Saphra and Lopez, 2019b; Murty et al., 2023; Chen et al., 2024; Merrill et al., 2023). MI still strongly builds on the circuit paradigm that operates at the level of module interactions (Lieberum et al., 2023; Marks et al., 2024; Merullo et al., 2024; Tigges et al., 2024; Hanna et al., 2024; Dunefsky et al., 2024)—a framework which also inspires NLPI researchers (Ferrando et al., 2024; Ferrando and Voita, 2024). Work from Anthropic often becomes an MI focus, such as when promising results using SAEs (Bricken et al., 2023) inspired a flurry of followup work (Templeton et al., 2024; Gao et al., 2024; Lieberum et al., 2024; Belrose, 2024; Rajamanoharan et al., 2024a,b; Karvonen et al., 2024; Braun

et al., 2024; Kissane et al., 2024; Gorton, 2024; Makelov, 2024)—though sparse encoding is another longstanding interest of NLPI (Subramanian et al., 2018; Niculae et al., 2018; Panigrahi et al., 2019; Meister et al., 2021; Prouteau et al., 2022; De Cao et al., 2022; Guillot et al., 2023).

Fortunately, there are signs of increasing unity in scientific focus. Some academics connected to the MI community have promoted interest in tools from linguistics and cognitive science (Wang et al., 2023; Arora et al., 2024). Speaker lineups at MI meetings often include longstanding NLPI researchers (MIT, 2023; Bau et al., 2024; Barez et al., 2024). MI researchers have also begun to engage more deliberately with peer-reviewed general ML conferences (Nanda, 2023b), though this effort has not extended to the specialized NLPI tracks and venues that focus on similar objectives and methods.[13]

## 4   Conclusion

Whatever terminological confusion and ideological tension they have brought to the interpretability field, the MI community is also responsible for its newfound popularity. The interest, energy, and opportunities MI brings to the field cannot be understated, nor should they be taken for granted. NLPI and MI researchers alike are motivated by social responsibility, intellectual curiosity, and the possibility of improving our tools. However, many MI researchers are also members of the alignment community concerned about catastrophic AI risk, where the value of MI is questioned (Greenblatt et al., 2023; Kross, 2023; Segerie, 2023).

There may come a time when alignment community consensus turns away from MI. Though many current MI researchers may leave—and some generous resources could disappear—others are likely to continue pursuing our shared objectives. Our communities have too much in common: scientific curiosity and a belief that we should understand the tools we use. We will all continue striving for that objective as long as there are opaque models to understand. Why not, therefore, also aim to connect?

---

[10] Effective Altruist charities have distributed millions in MI research grants (Open Philanthropy; EA Grants; Future of Life Institute).

[11] Many elite institutions have student societies where existential risk and MI are discussed over meals (Washington Post, 2023).

[12] In addition to cultural differences around scientific practice, there are also differences in preferred venues. ACL and BlackBoxNLP have struggled to engage the MI community, who prefer ML venues and the creation of new workshops (Barez et al., 2024). Prof. Yonatan Belinkov of Technion, a BlackBoxNLP founder, posted a call for MI researchers to submit to ACL venues (Belinkov, 2023b) and BlackBoxNLP 2023 (Belinkov et al., 2023) attempted to bridge the gap by inviting MI researchers to participate in a panel, where this divide was discussed.

---

[13] A point conceded by Neel Nanda, a leading MI researcher (Belinkov et al., 2023). The ACL preprint policy was a discouraging factor, but this is fortunately no longer the case (ACL Executive Committee, 2024).

## Acknowledgments

We thank the authors of all tweets cited in this paper for granting us permission to reprint their tweets. We'd also like to thank (in alphabetical order): Fazl Barez, Yonatan Belinkov, Yanai Elazar, Thomas Fel, Neel Nanda, Chris Olah, Yuval Pinter, Ashish Sabharwal, Oyvind Tafjord, and the anonymous reviewers for engaging in discussion with us and providing valuable feedback. Thanks to Lelia Glass, Peter Hase, and Aryaman Arora for providing references.

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

## A Understanding language about understanding language models

The interpretability field struggled with terminological clarity and consensus long before *mechanistic* entered the lexicon (Doshi-Velez and Kim, 2017; Lipton, 2018; Rudin, 2019; Riedl, 2019; Jacovi and Goldberg, 2020). By even using the word *interpretability*, we implicitly dismiss the distinction drawn by Rudin (2019) between large "black-box" neural models and models which are designed to be understood: particularly, that the latter can be interpreted, but the former only explained.[14]

---

[14]As AI capabilities have advanced, this position against post-hoc black-box explanation has become less popular: Intrinsically interpretable models are often less performant (Madsen et al., 2024) and cannot always guarantee the human understanding that motivates their use (Lipton, 2018). As machine learning researchers have rejected the argument against black-box explanation (Jacovi and Goldberg, 2020), they have also abandoned any semantic distinction between explanation and interpretation.

While clarity is always important in scientific language, the nature of interpretability research makes it all the more urgent to speak precisely. As a community, we aim to understand the behavior of models and how they work, but how can we shed any light on these inner workings by leveraging confusing jargon? In fact, the ambiguity of *mechanistic* is emblematic of a wider struggle to communicate interpretability research effectively.

Let us consider some other sticking points in the interpretability lexicon. A core part of the "Is attention explanation?" debate (Jain and Wallace, 2019; Serrano and Smith, 2019; Wiegreffe and Pinter, 2019; Bibal et al., 2022; Jain and Wiegreffe, 2023) is a disagreement over whether an *explanation* must be *faithful* by definition (Wiegreffe and Pinter, 2019, sec. 5). Subsequent work (Wiegreffe and Pinter, 2019; Jacovi and Goldberg, 2020) delineated between faithful and *plausible* (Herman, 2017), or human-acceptable (Wiegreffe et al., 2022), explanation. Even the terminology used to describe the format of textual explanations has been a source of discussion and disagreement (Jacovi and Goldberg, 2021; Wiegreffe and Marasović, 2021)—such as whether "extractive" and "abstractive," terms borrowed from the summarization literature, adequately characterize the difference between types of textual explanations.

In the MI literature, there have been terminology overloads or semantic disagreements over words like *feature* and *illusion*. The term *feature* has been used to describe mono-semantic concept representations of neurons derived from SAEs (Mueller et al., 2024), though it is more widely and historically associated with vector representations of data (text) that are either manually designed ("feature engineering") or learned by neural networks (Bereska and Gavves, 2024). A debate about subspace activation patching has centered around the meaning of the word *illusion*, namely, whether it applies to any dimension that becomes clearly causally relevant only when its causal role is tested with an intervention (Makelov et al., 2024), or whether such artifacts are a natural—and even explanatory—product of the model's representational geometry, and therefore informative of its true structure (Wu et al., 2024).

All of these examples, however, center around the need to ground our empirical work in precise vocabulary—not, like *mechanistic interpretability*, around the designation of group identity (§3.2.2). Terminological disagreements are usually resolved through discourse in shared venues. The NLPI community's adoption of the term *mechanistic* did not follow the same pattern (§3.3); its use may give the impression of cohesion and unity, but it masks a deep division which leads to duplicated research efforts and limits shared knowledge. Such outcomes will only hinder progress towards our shared goal: more deeply understanding language models.