# OpenReview forum: "Mechanistic?"
_EMNLP/2024/Workshop/BlackBoxNLP — BlackboxNLP 2024_

### Official Review · Reviewer_otQ5 · 2024-09-09

**Overall Assessment:** 5
**Confidence:** 4

**Best Paper:**

1

**Best Paper Justification:**

-

**Comments Questions Suggestions And Typos:**

-

**Paper Summary:**

This paper explores the evolution and current state of "mechanistic interpretability" within the field of NLP and its corresponding research community. It provides a comprehensive overview of how this term has developed, its implications, and its reception by the traditional, academic NLP interpretability research community. The authors critically examine the divergence between traditional academic approaches and the emerging trends associated with mechanistic interpretability. They advocates for a more precise use of terminology and suggest strategies to foster better integration and dialogue between these two research communities with very similar interests.

**Summary Of Strengths:**

The paper provides a well-organized overview of the recent history of interpretability research, highlighting how the two communities in this field have developed in distinct directions. It offers a comprehensive examination of various perspectives, engaging with both academic literature and less formal sources like blog posts and tweets, which effectively capture the cultural context surrounding these developments. I think this paper is pretty unique in its type, and I believe that the subjects in it are worth discussing in the workshop.

**Summary Of Weaknesses:**

Readers who are not deeply familiar with both of the fields will find it hard to engage with. The paper tends to assume a certain level of prior knowledge that may not be accessible to all potential readers, which can limit its broader impact and accessibility. For example, the introduction uses the term "mechanistic interpretability community" is mentioned in the introduction without a definition before it. I think that the intro could be rewritten to offer a clearer, more accessible overview of interpretability research in general, followed by a specific explanation of mechanistic interpretability. While this paper is presented in the context of BlackboxNLP, I still believe it should still strive to be comprehensible to readers from outside these specialized fields or to newcomers.

The abstract mentions that the paper addresses four different angles, but it is not immediately clear what these four angles are. Clarifying this in the abstract would help set clearer expectations for the reader and provide a more structured overview of the paper's content.

---

### Official Review · Reviewer_gooX · 2024-09-11

**Overall Assessment:** 4
**Confidence:** 5

**Best Paper:**

1

**Best Paper Justification:**

N/A

**Comments Questions Suggestions And Typos:**

* There's perhaps a point that could be made that earlier deep learning papers were similar in many ways to interpretability papers, without the outright focus on causal and rigorous explanations (Erhan et al., 2010 -- first paper in missing citations below). I bring this up because it could perhaps be argued that the overemphasis on applications rather than understanding is a relatively recent and brief window in the history of deep learning research. That point is definitely starting to get outside of the main point of this paper, though, which is on the term "mechanistic". So I'll leave it up to the authors.

Missing citations:
* [Why Does Unsupervised Pre-training Help Deep Learning?
](https://www.jmlr.org/papers/volume11/erhan10a/erhan10a.pdf)
* [In-Context Learning Task Vectors](https://arxiv.org/abs/2310.15916) <-- Steering vectors (e.g., Ilharco et al.), also similar to below two
* [Function Vectors](https://arxiv.org/abs/2310.15213) <-- around L259, steering vectors
* [Language Models Implement Word2Vec](https://arxiv.org/abs/2305.16130) <-- same as above, L259
* [Universal Neurons in GPT-2](https://arxiv.org/abs/2401.121810) <-- for the section on neurons, probing, polysemanticity, etc.

**Paper Summary:**

This paper reflects on the recent surge interest in interpretability research, specifically in "mechanistic" interpretability. The authors provide perspective on what mechanistic means/used to mean/was intended to mean and describe both the cultural and technical senses of the word. The authors provide a history of interpretability research stemming from work in vision models, the early NLP interpretability community/work (NLPI), and finally the mechanistic interpretability research agenda. The authors point out that work that often claims to be mechanistic falls short of the narrow technical sense, which originates in psychology research, and instead uses the term to signal the use of any methods investigating the internals (weights or activations) of models, or signaling a cultural alignment with a specific sect of the AI safety community.

**Summary Of Strengths:**

* This paper is a timely and well researched take on the current state of interpretability. The history covered goes back to the beginnings of interpretability work and (I believe) accurately describes the rise of mechanistic interpretability.
* I think this work will be useful for those more familiar with the traditional (NLPI) community to better understand mechanistic interpretability, for those in the opposite camp (new to the field, less familiar with NLPI), and people unfamiliar with the field entirely. This is a perfect addition for BlackBox and I think will generate productive conversations. I left some suggestions below, but I would like to see the paper accepted.

**Summary Of Weaknesses:**

* For a position paper, this work is a bit lenient on a few stances of specific techniques that count as mechanistic or not (in the narrow definition). I understand that the authors do not want to call out any specific papers for being "not mechanistic", but I think it would strengthen the paper to take a stance on some of the techniques. Examples given in the paper are the logit lens and the often lacking causality analysis done on SAEs, but are path patching or causal mediation analysis properly mechanistic? The authors also mention the back and forth with Makelov, et al. and others without taking a position. I know this is a bit fraught, but given that the authors re-introduce a narrow technical definition of mechanistic, I think it would strengthen the paper to weigh in a bit here.

---

### Official Review · Reviewer_JrzA · 2024-09-11

**Overall Assessment:** 4
**Confidence:** 4

**Best Paper:**

1

**Best Paper Justification:**

No

**Comments Questions Suggestions And Typos:**

Interpretability works have also been motivated by the humanity (cogsci) field to connect neural nets to the theory of language (e.g., debate on the need for innate biases to achieve a certain language ability) [1][2], where the understanding of the tool (neural nets) is a prerequisite to impact a theory. Such a scientific view would also be worth further mentioning. Indeed, the paper [1] (Sec. 4) mentioned a “mechanistic” understanding of neural nets from the Cogsci perspective, although the term is just used as a contrast to behavioral understanding.

[1] Marco Baroni. "On the proper role of linguistically oriented deep net analysis in linguistic theorising." Algebraic structures in natural language. CRC Press, 2022. 1-16.
[2] Tal Linzen, Emmanuel Dupoux, and Yoav Goldberg. 2016. Assessing the Ability of LSTMs to Learn Syntax-Sensitive Dependencies. Transactions of the Association for Computational Linguistics, 4:521–535.

Perhaps this paper will cause negative feelings in the MI community and lead to further separation of the NLPI--MI communities, which will not be the authors' intention; such a hedge statement might be good in the limitation section.

**Paper Summary:**

This paper is focused on the definition of the jargon--- “mechanistic interpretability" ---recently used in the interpretability community. The authors first discussed the definitions of the term from multiple perspectives and then connected its scopes to relevant interdisciplinary contexts in NLP, psychology, and philosophy. The authors are especially concerned about the scarce connection between the traditional NLP interpretability (NLPI) community and the emerging mechanistic interpretability (MI) one in the latter part of the paper. The historical clash between these communities is objectively described, and an attempt is made to attract the attention of MI people to the (classical) NLPI works through this paper.

**Summary Of Strengths:**

This paper is well-written and a good snapshot of the situations the NLPI community is currently facing. Such a framing of the paper will fairly fit the BlackBoxNLP workshop. The citations are rich enough (I hope every NLPI/MI researcher should follow the terminologies in Section 4); I hope it will benefit both the NLPI and MI communities to catch up with each other's scientific histories.
The definition of the term is discussed from multiple perspectives; specifically, the cultural definition will be worth caring about in order to develop a healthy community.

**Summary Of Weaknesses:**

Objective evidence will further enhance the argument. For example, one can analyze the citation network to confirm whether both communities are truly dissociated. The mention that "the mechanistic interpretability community is shifting from vision to the NLP domain" (Sec. 3.2.2) is confirmed based on the Circuits thread, but the broader trends can also be tested by quantitatively analyzing the macro trend of the usage of the term, e.g., modeling the semantic change of the word “mechanistic interpretability" in CS papers, which may produce a nice illustration that the two communities indeed clashed.

---

### Decision · Program_Chairs · 2024-09-18

**Decision:**

Accept

**Comment:**

All reviewers agree that the paper is well-written and provides a comprehensive overview of mechanistic interpretability (MI), tracing its history and evolution, and discussing the divide between MI and traditional NLP interpretability (NLPI). This timely work is a valuable resource for both NLPI and MI communities, making it well-suited for the workshop